# Excess years of life lost to COVID-19 and other causes of death by sex, neighbourhood deprivation, and region in England and Wales during 2020: A registry-based study

Evangelos Kontopantelis[1,2,3]*, Mamas A. Mamas[1,4,5], Roger T. Webb[6,7], Ana Castro[8], Martin K. Rutter[9,10], Chris P. Gale[11,12,13], Darren M. Ashcroft[2,7,14], Matthias Pierce[15], Kathryn M. Abel[15], Gareth Price[16], Corinne Faivre-Finn[16], Harriette G. C. Van Spall[17], Michelle M. Graham[18], Marcello Morciano[2,3], Glen P. Martin[1], Matt Sutton[2,19], Tim Doran[8]

1 Division of Informatics, Imaging and Data Sciences, University of Manchester, Manchester, England, 2 NIHR School for Primary Care Research, University of Oxford, Oxford, England, 3 Health Organisation, Policy and Economics (HOPE) Research Group, University of Manchester, Manchester, England, 4 Keele Cardiovascular Research Group, Centre for Prognosis Research, Keele University, Keele, England, 5 Department of Cardiology, Jefferson University, Philadelphia, Pennsylvania, United States of America, 6 Centre for Mental Health & Safety, Division of Psychology & Mental Health, University of Manchester and Manchester Academic Health Sciences Centre (MAHSC), England, 7 NIHR Greater Manchester Patient Safety Translational Research Centre, Manchester, England, 8 Department of Health Sciences, University of York, England, 9 Division of Diabetes, Endocrinology and Gastroenterology, School of Medical Sciences, University of Manchester, Manchester, England, 10 Diabetes, Endocrinology and Metabolism Centre, Manchester University NHS Foundation Trust, Manchester Academic Health Sciences Centre, Manchester, England, 11 Leeds Institute of Cardiovascular and Metabolic Medicine, University of Leeds, Leeds, England, 12 Leeds Institute for Data Analytics, University of Leeds, Leeds, England, 13 Department of Cardiology, Leeds Teaching Hospitals NHS Trust, Leeds, England, 14 Division of Pharmacy & Optometry, University of Manchester, Manchester, England, 15 Centre for Women's Mental Health, Division of Psychology and Mental Health, University of Manchester, Manchester, England, 16 Manchester Cancer Research Centre, The Christie NHS Foundation Trust, University of Manchester, Manchester, England, 17 Department of Medicine and Department of Health Research Methods, Evidence, and Impact, McMaster University, Hamilton, Canada, 18 Division of Cardiology, University of Alberta and Mazankowski Alberta Heart Institute, Edmonton, Alberta, Canada, 19 Division of Population Health, Health Services Research & Primary Care, University of Manchester, Manchester, England

* e.kontopantelis@manchester.ac.uk

**Data Availability Statement:** ONS weekly mortality data are available from https://www.ons.gov.uk/

## Abstract

### Background

Deaths in the first year of the Coronavirus Disease 2019 (COVID-19) pandemic in England and Wales were unevenly distributed socioeconomically and geographically. However, the full scale of inequalities may have been underestimated to date, as most measures of excess mortality do not adequately account for varying age profiles of deaths between social groups. We measured years of life lost (YLL) attributable to the pandemic, directly or indirectly, comparing mortality across geographic and socioeconomic groups.

### Methods and findings

We used national mortality registers in England and Wales, from 27 December 2014 until 25 December 2020, covering 3,265,937 deaths. YLLs (main outcome) were calculated using

peoplepopulationandcommunity/birthsdeathsand
marriages/deths/datasets/weeklyprovisionalfigure
sondeathsregisteredinenglandandwales. However,
the stratification of the data used in the analyses is
not freely available and can be obtained from the
ONS (health.data@ons.gov.uk).

**Funding:** The authors received no specific funding
for this work.

**Competing interests:** I have read the journal's
policy and the authors of this manuscript have the
following competing interests: CG declares
receiving support from Abbot and BMS towards
this work, consulting fees from Amgen and
AstraZeneca, honoraria from AstraZeneca,
participation in Data Safety Monitoring Boards for
several trials (TACTIC, DUAL-ACS, DANBLOCK,
PROFID, RAPID NSTEMI, STEEER-AF), stock
options with the European Heart Journal Quality of
Care and Clinical Outcomes as Deputy Editor, and
other financial interests with Wondr Medical. MKR
declares Cell catapult and NovoNordisk consulting
fees, unrelated to the content of the manuscript.
None of the other authors have anything relevant to
declare.

**Abbreviations:** ASMR, age-standardised mortality
rate; COVID-19, Coronavirus Disease 2019; IMD,
Index of Multiple Deprivation; LSOA, Lower Super
Output Area; NHS, National Health Service; ONS,
Office for National Statistics; WHO, World Health
Organization; YLL, years of life lost.

2019 single year sex-specific life tables for England and Wales. Interrupted time-series analyses, with panel time-series models, were used to estimate expected YLL by sex, geographical region, and deprivation quintile between 7 March 2020 and 25 December 2020 by cause: direct deaths (COVID-19 and other respiratory diseases), cardiovascular disease and diabetes, cancer, and other indirect deaths (all other causes). Excess YLL during the pandemic period were calculated by subtracting observed from expected values. Additional analyses focused on excess deaths for region and deprivation strata, by age-group. Between 7 March 2020 and 25 December 2020, there were an estimated 763,550 (95% CI: 696,826 to 830,273) excess YLL in England and Wales, equivalent to a 15% (95% CI: 14 to 16) increase in YLL compared to the equivalent time period in 2019. There was a strong deprivation gradient in all-cause excess YLL, with rates per 100,000 population ranging from 916 (95% CI: 820 to 1,012) for the least deprived quintile to 1,645 (95% CI: 1,472 to 1,819) for the most deprived. The differences in excess YLL between deprivation quintiles were greatest in younger age groups; for all-cause deaths, a mean of 9.1 years per death (95% CI: 8.2 to 10.0) were lost in the least deprived quintile, compared to 10.8 (95% CI: 10.0 to 11.6) in the most deprived; for COVID-19 and other respiratory deaths, a mean of 8.9 years per death (95% CI: 8.7 to 9.1) were lost in the least deprived quintile, compared to 11.2 (95% CI: 11.0 to 11.5) in the most deprived. For all-cause mortality, estimated deaths in the most deprived compared to the most affluent areas were much higher in younger age groups, but similar for those aged 85 or over. There was marked variability in both all-cause and direct excess YLL by region, with the highest rates in the North West. Limitations include the quasi-experimental nature of the research design and the requirement for accurate and timely recording.

## Conclusions

In this study, we observed strong socioeconomic and geographical health inequalities in YLL, during the first calendar year of the COVID-19 pandemic. These were in line with long-standing existing inequalities in England and Wales, with the most deprived areas reporting the largest numbers in potential YLL.

## Author summary

### Why was this study done?

- The Coronavirus Disease 2019 (COVID-19) pandemic generated large numbers of excess deaths (additional deaths over those predicted from trends in previous years). These excess deaths were also unevenly distributed across different geographic areas and socioeconomic groups, exacerbating prepandemic inequalities.

- Up to a quarter of the excess deaths during the pandemic were attributable to coronary heart disease, dementia, and other noninfectious causes, and not directly to COVID-19 infection.

- Most measures of excess deaths do not fully account for differences in the ages at which people die in different social groups. If the pandemic killed disproportionally more

young people in more deprived areas, then inequalities resulting from the pandemic will have been underestimated. Measuring years of life lost (YLL) rather than excess deaths would address this problem.

## What did the researchers do and find?

- In England and Wales, there were an estimated 763,550 (95% CI: 696,826 to 830,273) excess YLL during the first 42 weeks of the pandemic, of which 15% were not directly attributed to COVID-19 or another respiratory cause.

- For all-cause mortality, estimated deaths in the most deprived compared to the most affluent areas were as follows: 11 times as many for 15- to 44-year-olds, 3 times as many for 45- to 64-year-olds, 40% higher in 75- to 84-year-olds, and not significantly higher for those aged 85 or over.

- This pattern of disproportionately higher mortality in younger age groups exacerbated prepandemic inequalities between the most and least deprived areas, and varied widely across regions, with the North West particularly affected.

## What do these findings mean?

- Inequalities between socioeconomic and geographic groups resulting from the COVID-19 pandemic are more pronounced than previously reported.

- Future plans to manage pandemics, including decisions about vaccination rollout, should include an understanding of regional and socioeconomic variation in YLL and how this has exacerbated long-standing health inequalities.

- Immediate and longer-term recovery planning for communities and their health and social services should reflect historical disparities as well as the impact of the pandemic on YLL.

- Limitations of this study include the observational nature of the data and the need for accurate and timely recording of the deaths and their causes.

## Introduction

Estimating excess deaths, by comparing observed deaths with the number expected based on historical trends, is an informative way of quantifying the potential impact of a pandemic on different population groups. There were an estimated 126,658 excess deaths in England in the first year of the Coronavirus Disease 2019 (COVID-19) pandemic [1], and these deaths were unevenly distributed both socioeconomically and geographically. Socioeconomic factors underlie much of the geographic variation, mediating their effects through population health status, living and work circumstances, environmental conditions, and infection prevention and mitigation measures. Although all English regions and all social groups have been detrimentally affected by the pandemic, age-standardised rates in the first wave were 67% higher

in the West Midlands than in the South West, and 33% higher in the most deprived than the most affluent quintile [2].

The count of all-cause excess deaths includes those directly attributable to the virus and additional indirect deaths resulting from public health measures and the wider societal response, which for some causes will be lower than predicted. In the early part of the current pandemic, a quarter of deaths were attributable to causes other than COVID-19 infection, in particular cardiovascular disease and dementia [1]. These indirect excess deaths followed different regional and socioeconomic patterns depending on underlying cause [2], and resulted from changes in primary and secondary care capacity [3,4], and in social behaviour, including decisions on seeking care [5,6], resulting in increases in mortality for conditions such as cardiovascular disease and cancer [7–9]. In previous work, we quantified excess deaths in the first 30 weeks of the pandemic for a range of causes [2]. A quarter of all excess deaths in England and Wales were not directly attributable to COVID-19 infection, which aligned with similar reports from the US, Brazil, Italy, and Germany [7–10]. Socioeconomic patterns in excess deaths differed by underlying cause, with a strong socioeconomic gradient observed for respiratory, "other" indirect causes, and any cause. The most deprived fifth of areas reported 124 (95% CI: 121 to 127) any-cause excess deaths per 100,000 population compared to 93 (95% CI: 90 to 96) for the least deprived fifth. However, there was no clear gradient for excess deaths from cardiovascular disease and diabetes, nor cancer [2].

The risk of dying following infection with COVID-19 increases sharply with age [11]. However, susceptibility in younger age groups may also vary across social groups, giving rise to inequalities that are not adequately captured by a simple count of deaths. Additionally, indirect deaths are unlikely to follow the same age-related patterns as deaths directly attributable to COVID-19 and may result in substantial numbers of deaths in younger age groups. Years of life lost (YLL) is an alternative measure of premature mortality that allows for comparisons between causes of death and across population groups, because it accounts for both the number of deaths and the age at which those deaths occurred (being higher for deaths that occur at a younger age).

In this study, we measure YLL attributable to the pandemic by cause of death, comparing mortality across sexes and geographic and socioeconomic groups in England and Wales. We hypothesised that more deprived areas would not only have higher excess deaths, but the age distribution of these deaths would differ from more affluent areas, with more younger people dying, directly or indirectly, during the pandemic. To the best of our knowledge, this aspect of the pandemic has not been quantified previously for England and Wales.

## Methods

### Data

Deaths registered in England and Wales from 27 December 2014 until 1 March 2021 were accessed through the secure Amazon Workspaces environment, operated by NHS Digital. NHS Digital is the national provider of information, data, and IT systems within England's National Health Service (NHS). To minimise the effect of delayed death registration, the observation period was from the first week of 2015 (ending 2 January) to the last week of 2020 (ending 25 December). For each death, the underlying cause and up to 15 contributory causes are recorded on the death certificate. The underlying cause is defined as the disease or injury that initiated the series of events leading directly to death, or the circumstances of the accident or violence that produced the fatal injury [12]. In addition, information is also available on the following: unique NHS identifier, age, sex, registered date of death, residential postcode, and place of death category. For each death, potential YLL were calculated using 2019 single year

sex-specific life tables for England and Wales [13,14]. Those aged over 100 were assumed to have the life expectancy of centenarians of the same sex.

Using the postcode of residence for each case, the database was linked to Lower Super Output Areas (LSOAs), which are small geographical areas containing a median of around 1,500 residents. Additional information was linked at the LSOA level, including region, population sizes by age and sex strata, and area-level socioeconomic deprivation information. Deprivation was measured using the Index of Multiple Deprivation (IMD), a composite score across 7 domains: income, employment, health, education and skills, housing, crime, and environment. Deprivation rankings and scores often remain relatively constant over time [15], and so we used the latest available IMDs reported as scores (for 2019 in England and 2014 in Wales). Population estimates at the LSOA level were available from the Office for National Statistics (ONS) up to 2019 [16] and were extrapolated to 2020 using simple linear regression, with year modelled as a continuous predictor. Regional information was available at a high level, that of the 10 former Strategic Health Authorities in England and Wales, which were responsible for management of performance, enacting directives and implementation of health policy locally as required by the Department of Health and Social Care.

Deaths were organised in 2 main categories: direct or indirect. Direct deaths were those where the underlying cause of death was attributed to COVID-19 (ICD-10 code U071 virus identified, or U072 virus not identified), plus respiratory deaths (J00–J99). This aggregation was used because coding of COVID-19 varied over the study's observation period, with an unknown number of COVID-19 deaths being attributed to other respiratory diseases, especially in the early stages of the pandemic and in care homes [17]. For comparison, we also present and model non-COVID-19 respiratory deaths separately. We examined 3 subcategories of deaths indirectly related to COVID-19: cardiovascular and diabetes (ICD-10 codes: I00–I99 and E10-E14, except for I426 alcoholic cardiomyopathy), since these are closely related disorders; cancer (ICD-10 codes: C00–D48); and other indirect deaths (including drug-related and alcohol-specific deaths, suicides, accidents, and all other causes). These categories were selected to reflect emerging evidence on increased mortality rates for these specific causes during the pandemic [5,7–9], but also reflected feasibility concerns. In particular, that conditions should be prevalent enough across age groups to be modelled reliably, but also that the selected categories are robust to time lags in coronial verdicts. The "other indirect" deaths category included deaths where the underlying cause was still under investigation by the coroner, which in most cases are external causes. Each subcategory was modelled separately. Categories were mutually exclusive, so a few deaths that could be attributed to 2 categories were not double counted. For example, cardiovascular deaths did not include alcoholic cardiomyopathy, which was included in the "other indirect" category. "Other" deaths include a very small number of records where there was no underlying cause of death, or where the cause was being further investigated. The study was conducted using aggregated national mortality data; ethical approval was, therefore, not needed. This study is reported as per the Strengthening the Reporting of Observational Studies in Epidemiology (STROBE) guideline (S1 Checklist).

## Analyses

Data were imported, cleaned, and formatted as a time series in Stata v16. Total YLL for each category were aggregated, initially without stratification and then across the following strata: by sex, by region, and by deprivation quintile (1 = least deprived; 5 = most deprived). Sex-specific age-standardised mortality rates (ASMRs) per 100,000 population (and 95% confidence intervals) were computed using the World Health Organization (WHO) World Standard

Population to allow for global comparisons [18] and are presented graphically across the examined strata. Aggregated datasets were extracted from the secure environment and analysed locally. The study did not have a prospective protocol or analysis plan.

Time series of aggregate YLL for each examined death category were analysed in similar models. To model YLL, data from week 9 in 2020 (22 to 28 February), 2 weeks before the first officially confirmed COVID-19 related deaths (7 to 13 March), were set to "missing". A linear regression model with Newey–West standard errors was used on the aggregated England and Wales data, while linear regression models with Driscoll–Kray standard errors were used for all the panel time series (with both error structures allowing for time autocorrelation, with the latter also allowing correlation between panels) [19]. We used the built-in *newey* and the user-written *xtscc* time-series analysis commands in Stata [20], which produce standard errors that account for autocorrelation, with the maximum lag order for the autocorrelation structure being set to 52 weeks (a year, since we expected annual seasonality in the time series). As well as fitting time as a continuous covariate, the models included week number (1 to 52) as a categorical covariate, to account for potential seasonality and to obtain more accurate predictions for the investigated time period. Each of the panel time series models included a categorical term of the respective panel (i.e., sex, region, or deprivation quintile). An additional model included region, deprivation quintile, and their interaction term. For weeks 11 (7 to 13 March) to 52 (19 to 25 Dec 2020), we used the *margins* postestimation command to obtain the linear prediction from each model (and its standard error, computed using the delta method), which provides an estimate of the weekly expected YLL. This estimate was then subtracted from the respective weekly observed YLL to give an estimate of weekly excess YLL. To better quantify the YLL respective to the size of the population, we also computed YLL per 100,000 population and repeated all analyses. Additional figures by each categorisation of interest are provided in S1 Appendix. We also provide the sex-specific ASMRs in data files to enable international comparisons. Our previously reported analyses on excess all-cause and direct deaths were updated for this study period, allowing us to quantify excess mortality across deprivation quintiles and regions, by age group [21]. This allowed us to quantify the YLL per excess death and report excess number of deaths across age groups. These interactions were not analysed in other death categories due to low cell counts. We also examined the performance of the models by using 2015–2018 data to predict YLLs in 2019, expecting no excess YLLs, as was the case.

## Results

### All-cause

There were 3,265,937 deaths during the study period, with 486,796 deaths during the pandemic period in 2020 (7 March to 25 Dec). We estimated 763,550 (95% CI: 696,826 to 830,273) excess YLL during the study period across England and Wales, equivalent to 15% (95% CI: 14 to 16) of all-cause YLL observed during the equivalent time period in 2019. Excess YLL were higher for males (461,919; 95% CI: 426,408 to 497,430) compared to females (301,631; 95% CI: 267,557 to 335,706) and were highest in London (126,761; 95% CI: 121,370 to 132,152) and the North West (121,017; 95% CI: 110,647 to 131,387), and lowest in the South Central (39,953; 95% CI: 33,469 to 46,436) and the South West of England (35,032; 95% CI: 27,142 to 42,923). Excess YLL were lowest in the least deprived quintile (118,457; 95% CI: 106,554 to 130,360) and highest in the most deprived quintile (220,798; 95% CI: 204,705 to 236,891). This inequality is also captured in the sex-specific ASMRs, across both sexes (Fig 1). Excess all-cause deaths across age groups, by deprivation quintile and region, are presented in Table E in S1 Appendix, with a larger ratio of excess deaths in the most deprived over the least deprived areas in

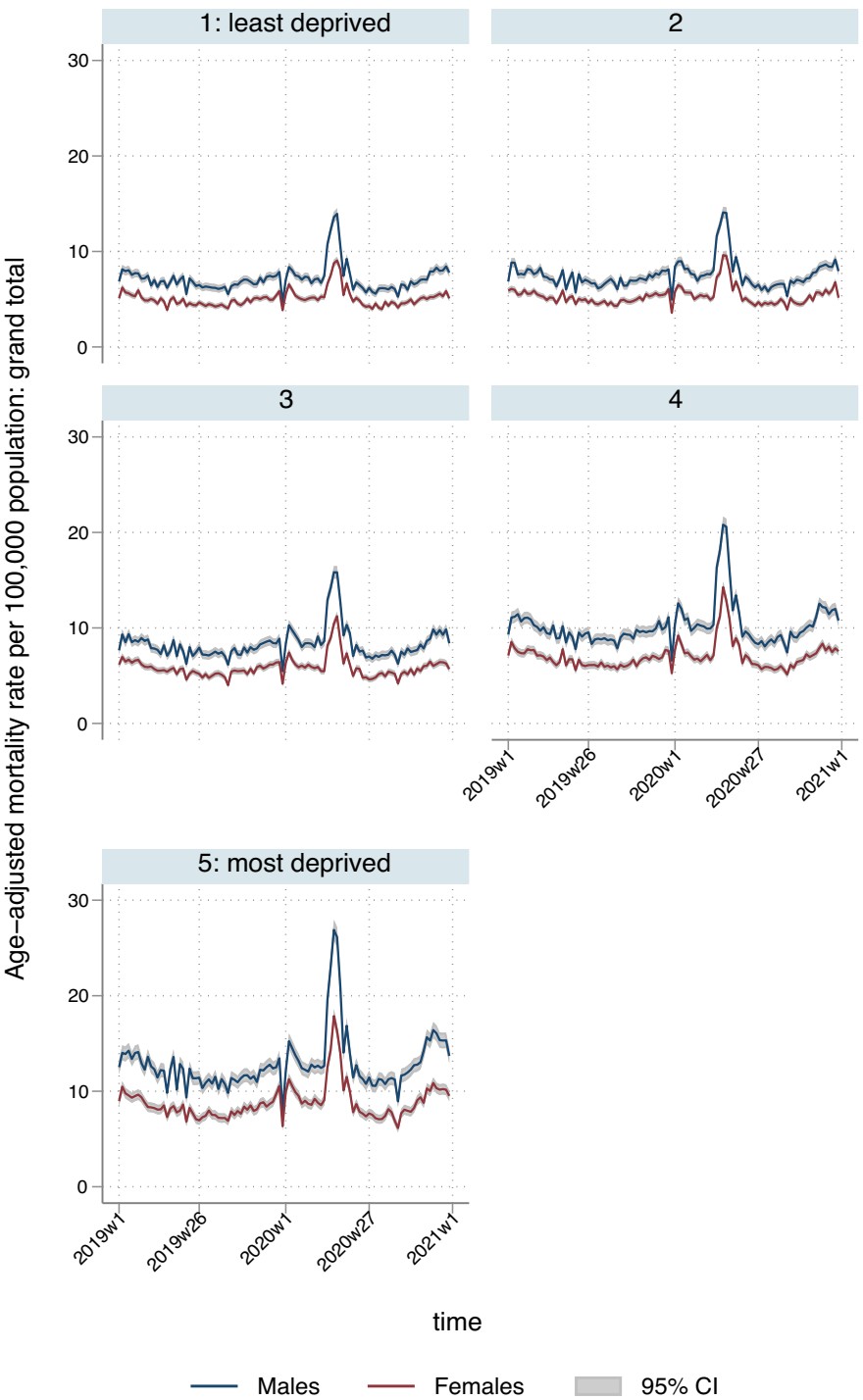

**Fig 1. Temporal patterns in ASMRs per 100,000 population for all deaths, by sex and deprivation quintile.** ASMR, age-standardised mortality rate.

younger age groups. For ages 45 to 64: 3.00 (95% CI: 2.34 to 3.67); 65 to 74: 1.93 (95% CI: 1.51 to 2.34); 75 to 84: 1.38 (95% CI: 1.14 to 1.62); 85+: 0.95 (95% CI: 0.76 to 1.14). The North West and London had the highest number of excess deaths in people aged 74 or below. Interacting

region and deprivation quintile revealed socioeconomic trends that varied by region, with the North West in particular, but also the West Midlands, reporting many more YLLs in areas corresponding to the most deprived quintile, nationally.

There were 1,125 (95% CI: 997 to 1,252) excess YLL per 100,000 nationally, ranging from 490 (95% CI: 319 to 661) per 100,000 in the South West to 1,550 (95% CI: 1,384 to 1,716) per 100,000 in the North West. Excess YLL per 100,000 population were unequally distributed across deprivation quintiles, with similar rates in quintiles 1 (916; 95% CI: 820 to 1,012) to 3 (977; 95% CI: 869 to 1,085), but sharply increasing for quintiles 4 (1,218; 95% CI: 1,089 to 1,346) and 5 (1,645; 95% CI: 1,472 to 1,819). Patterns were broadly similar to those for deaths caused directly by COVID-19 or other respiratory causes, which comprised the majority of excess deaths over the study period (Fig 2). Excess all-cause deaths across age groups per 100,000 population, by deprivation quintile and region, are presented in Table F in S1 Appendix, with a similar pattern of higher excess deaths for younger people in more deprived areas, and more pronounced for the older age groups, due to a well-known socioeconomic gap between the young and the old [22]. For ages 45 to 64: 2.26 (95% CI: 1.75 to 2.78); 65 to 74: 1.75 (95% CI: 1.37 to 2.13); 75 to 84: 2.58 (95% CI: 1.98 to 3.15); 85+: 1.63 (95% CI: 1.26 to 2.00). Assuming estimated excess all-cause YLL were only distributed among estimated excess all-cause deaths, a mean of 9.1 years per death (95% CI: 8.2 to 10.0) were lost in the least deprived quintile, compared to 10.8 (95% CI: 10.0 to 11.6) in the most deprived. The North West and London continued to have the highest rate of excess deaths, closely followed by West Midlands.

## Respiratory disease

Between 7 March 2020 and 25 December 2020, there were an estimated 646,518 (95% CI: 632,925 to 660,111) excess YLL in England and Wales attributed to COVID-19 or another underlying respiratory cause (direct, Table 1), representing an 123% increase (95% CI: 120% to 126%) compared to the equivalent time period in 2019 (underlying respiratory deaths alone). Of these, 387,406 (95% CI: 380,649 to 394,163) were in males and 259,112 (95% CI: 251,991 to 266,233) in females. There was a steep deprivation gradient, ranging from 92,782 (95% CI: 90,595 to 94,968) excess YLL in the least deprived quintile to 181,298 (95% CI: 177,509 to 185,086) in the most deprived. This inequality is reflected in the sex-specific ASMRs, with rates for males in the most deprived areas approximately double those of females in the most deprived areas or of males in the least deprived areas (Fig 3). Excess respiratory deaths across age groups, by deprivation quintile and region, are presented in Table E in S1 Appendix, with a larger ratio of excess deaths in the most deprived over the least deprived areas in younger age groups (for ages 45 to 64: 2.95 (95% CI: 2.79 to 3.10); 65 to 74: 2.02 (95% CI: 1.89 to 2.14); 75 to 84: 1.39 (95% CI: 1.32 to 1.46); 85+: 0.94 (95% CI: 0.89 to 0.99)). The North West and London had the highest numbers of excess respiratory deaths for people aged 74 or below. Interacting region and deprivation quintile revealed socioeconomic trends that varied by region and were similar to the patterns observed for all-cause deaths, with the North West reporting the largest excess YLLs spike in areas categorised in the most deprived quintile nationally.

The overall excess YLL for England and Wales directly attributed to COVID-19 infection or respiratory causes was 1,066 (95% CI: 1,041 to 1,091) per 100,000 population (Table 2), but there was substantial regional variation: rates ranged from 438 (95% CI: 408 to 468) per 100,000 in the South West of England to 1,529 (95% CI: 1,494 to 1,564) per 100,000 in the North West. For males, excess YLL per 100,000 population were 1,293 (95% CI: 1,269 to 1,317) for males and, for females, 844 (95% CI: 817 to 870). In terms of area-level deprivation, rates

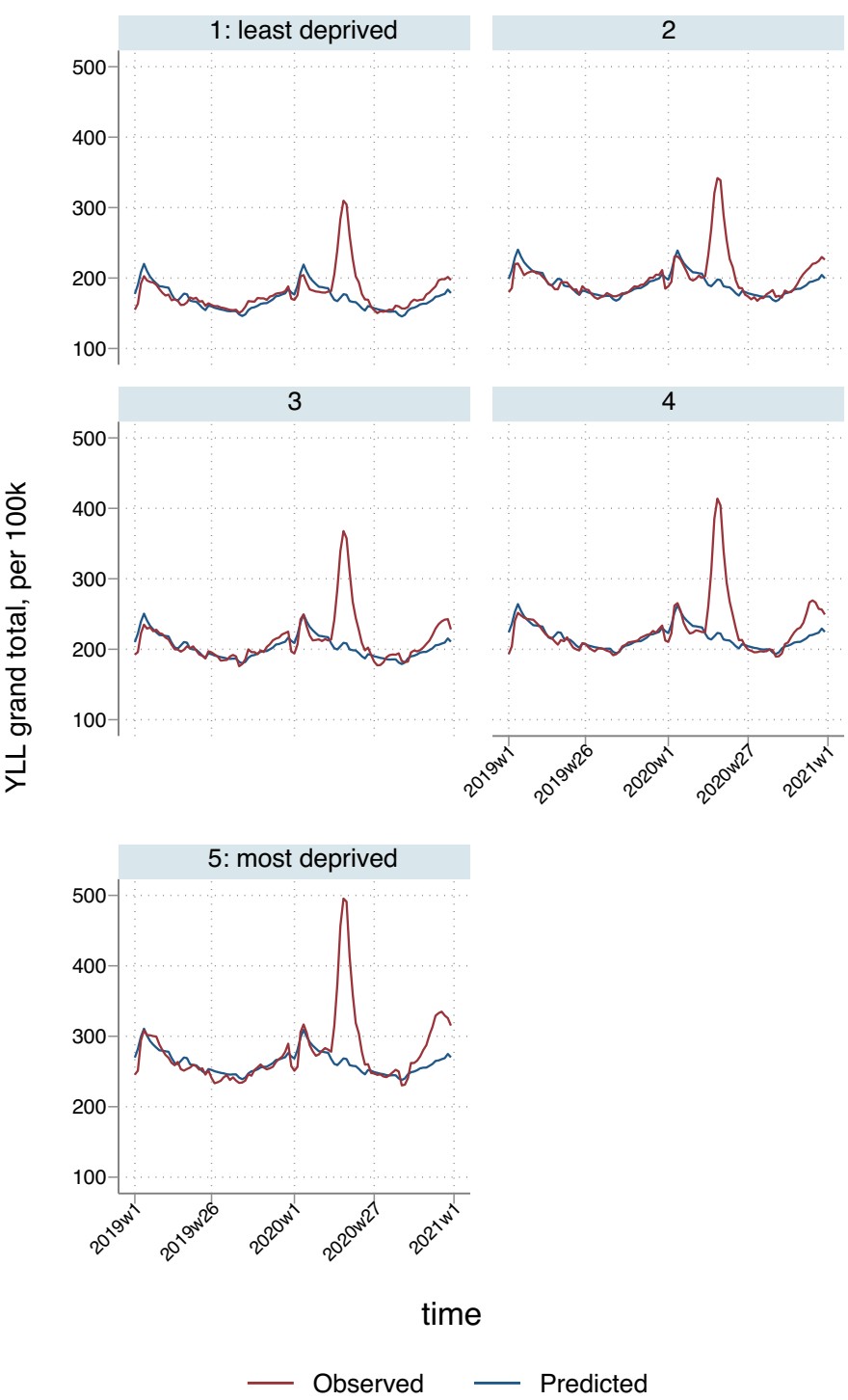

**Fig 2. Temporal patterns in YLL per 100,000 population due to all-cause deaths, by deprivation quintile**\*. \*3-week smoothed. YLL, years of life lost.

ranged from 789 (95% CI: 770 to 809) per 100,000 for the least deprived quintile to 1,493 (95% CI: 1,459 to 1,527) per 100,000 for the most deprived, reflecting higher spikes in more deprived regions and localities during the first and second phases of the pandemic (April to July and

**Table 1. Estimated excess YLL (95% confidence intervals), of direct, indirect, and total excess deaths, weeks 11 to 52, 7 March 2020 to 25 December 2020.**

| | Direct | | Indirect | | | |
| --- | --- | --- | --- | --- | --- | --- |
| | COVID or other respiratory | Other respiratory only | Cardiovascular and diabetes | Cancer | Other indirect deaths | Any cause |
| England and Wales | 646,518 (632,925, 660,111) | −77,542 (−91,135, −63,948) | 65,941 (47,542, 84,340) | −16,738 (−39,261, 5,785) | 67,828 (25,952, 109,705) | 763,550 (696,826, 830,273) |
| *Sex* | | | | | | |
| Males | 387,406 (380,649, 394,163) | −27,528 (−34,285, −20,771) | 50,877 (43,841, 57,913) | −10,967 (−21,769, −165) | 34,602 (10,012, 59,193) | 461,919 (426,408, 497,430) |
| Females | 259,112 (251,991, 266,233) | −50,014 (−57,134, −42,893) | 15,064 (2,610, 27,519) | −5,771 (−16,629, 5,087) | 33,226 (15,723, 50,729) | 301,631 (267,557, 335,706) |
| *Regions* | | | | | | |
| North East | 35,928 (34,767, 37,088) | −3,805 (−4,965, −2,644) | 4,495 (3,361, 5,629) | −512 (−1,971, 947) | 12,653 (10,181, 15,126) | 52,564 (49,050, 56,078) |
| North West | 111,972 (109,288, 114,656) | −14,408 (−17,092, −11,725) | 6,446 (3,932, 8,960) | −2,839 (−5,772, 95) | 5,438 (258, 10,618) | 121,017 (110,647, 131,387) |
| Yorkshire and Humber | 70,256 (68,890, 71,622) | −8,213 (−9,579, −6,847) | 3,237 (1,071, 5,404) | 455 (−2,072, 2,982) | 6,824 (1,731, 11,918) | 80,773 (75,667, 85,878) |
| East Midlands | 53,653 (52,370, 54,936) | −6,155 (−7,438, −4,872) | 8,098 (6,790, 9,406) | 1,079 (−1,048, 3,206) | 6,430 (1,630, 11,231) | 69,260 (62,816, 75,704) |
| West Midlands | 76,091 (74,534, 77,648) | −7,109 (−8,666, −5,552) | 11,563 (9,181, 13,945) | −375 (−2,492, 1,741) | 9,131 (5,511, 12,751) | 96,410 (89,036, 103,783) |
| East of England | 52,314 (50,868, 53,760) | −8,781 (−10,226, −7,335) | 2,069 (14, 4,125) | −6,723 (−9,590, −3,857) | 7,127 (280, 13,973) | 54,787 (45,868, 63,705) |
| London | 113,165 (111,765, 114,566) | −5,166 (−6,567, −3,766) | 10,052 (8,277, 11,827) | −3,423 (−5,615, −1,231) | 6,967 (2,869, 11,065) | 126,761 (121,370, 132,152) |
| South East Coast | 39,995 (38,837, 41,154) | −5,764 (−6,923, −4,606) | 5,186 (3,535, 6,836) | −3,251 (−5,220, −1,282) | 11,778 (8,239, 15,318) | 53,708 (48,555, 58,862) |
| South Central | 32,321 (31,073, 33,570) | −2,664 (−3,913, −1,416) | 6,165 (4,357, 7,973) | −1,267 (−2,968, 433) | 2,733 (−923, 6,390) | 39,953 (33,469, 46,436) |
| South West | 25,050 (23,607, 26,493) | −8,754 (−10,197, −7,310) | 5,369 (2,970, 7,768) | 1,128 (−1,730, 3,986) | 3,485 (−434, 7,405) | 35,032 (27,142, 42,923) |
| Wales | 35,012 (34,036, 35,987) | −6,092 (−7,068, −5,117) | 6,710 (5,187, 8,232) | 988 (−669, 2,645) | −608 (−2,864, 1,648) | 42,101 (38,114, 46,089) |
| *Deprivation quintiles* | | | | | | |
| 1 (least deprived) | 92,782 (90,595, 94,968) | −9,202 (−11,389, −7,016) | 11,972 (8,994, 14,950) | 1,994 (−2,104, 6,091) | 11,710 (2,694, 20,725) | 118,457 (106,554, 130,360) |
| 2 | 109,690 (106,955, 112,424) | −11,996 (−14,731, −9,262) | 6,573 (2,931, 10,216) | −4,081 (−8,680, 519) | 14,064 (6,488, 21,640) | 126,246 (113,665, 138,828) |
| 3 | 117,466 (115,136, 119,797) | −15,998 (−18,328, −13,667) | 13,412 (8,983, 17,841) | −7,339 (−10,882, −3,796) | 14,509 (5,813, 23,206) | 138,049 (126,893, 149,205) |
| 4 | 144,521 (141,710, 147,333) | −19,760 (−22,571, −16,948) | 14,657 (11,368, 17,947) | −3,742 (−8,442, 957) | 13,379 (5,847, 20,911) | 168,815 (156,103, 181,528) |
| 5 (most deprived) | 181,298 (177,509, 185,086) | −19,956 (−23,744, −16,167) | 22,776 (18,788, 26,764) | −1,573 (−7,053, 3,907) | 18,298 (10,785, 25,810) | 220,798 (204,705, 236,891) |

YLL, years of life lost.

November to December 2020; Figs 3 and 4). In addition to having higher baseline YLL, areas in the most deprived quintiles experienced the highest proportional increases in YLL during the pandemic; there was an 18% increase in the most deprived quintile compared to 15% in the least deprived (Table D in S1 Appendix). Per 100,000 population, ratios of excess deaths in the most deprived over the least deprived quintile were higher than in the population-unadjusted analyses, reflecting the differences in age distributions across deprivation strata, with

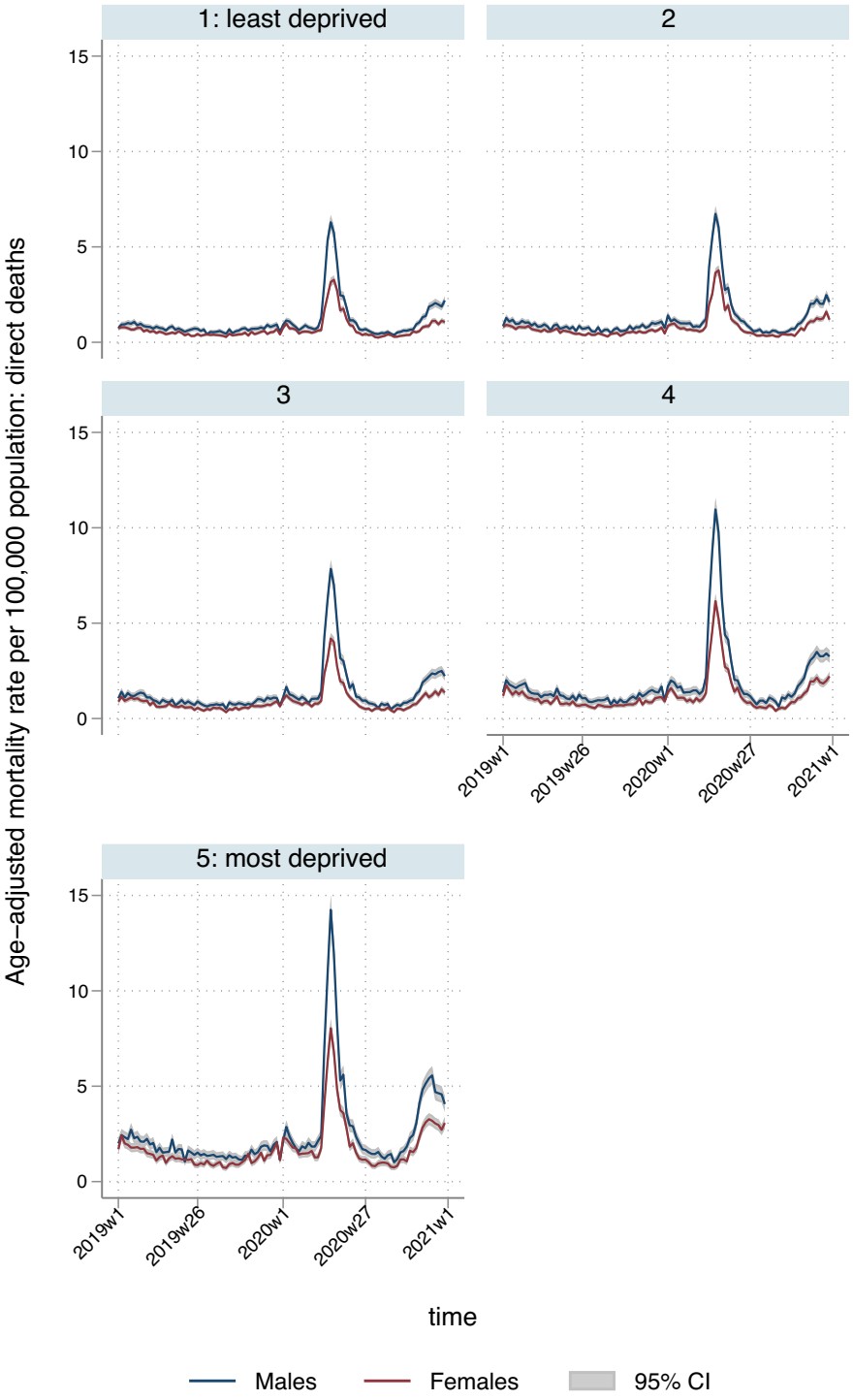

**Fig 3. Temporal patterns in ASMRs per 100,000 population for direct deaths, by sex and deprivation quintile.**
ASMR, age-standardised mortality rate.

older people more prevalent in more affluent locations [22]. For ages 45 to 64: 3.31 (95% CI: 2.99 to 3.65); 65 to 74: 2.86 (95% CI: 2.65 to 3.07); 75 to 84: 2.42 (95% CI: 2.27 to 2.59); 85+: 1.63 (95% CI: 1.53 to 1.74). Assuming estimated excess respiratory YLL were only distributed

**Table 2. Estimated excess YLL (95% confidence intervals), of direct, indirect, and total excess deaths per 100,000 population, weeks 11 to 52, 7 March 2020 to 25 December 2020.**

| | Direct | | Indirect | | | |
|---|---|---|---|---|---|---|
| | COVID or other respiratory | Other respiratory only | Cardiovascular and diabetes | Cancer | Other indirect deaths | Any cause |
| England and Wales | 1,066 (1,041, 1,091) | −146 (−171, −120) | 75 (31, 119) | −79 (−131, −27) | 63 (6, 119) | 1,125 (997, 1,252) |
| *Sex* | | | | | | |
| Males | 1,293 (1,269, 1,317) | −111 (−135, −87) | 128 (95, 162) | −92 (−143, −41) | 58 (−6, 123) | 1,387 (1,276, 1,499) |
| Females | 844 (817, 870) | −179 (−206, −153) | 23 (−30, 75) | −66 (−115, −18) | 66 (13, 118) | 866 (724, 1,007) |
| *Regions* | | | | | | |
| North East | 1,214 (1,174, 1,253) | −268 (−307, −228) | 55 (21, 88) | −98 (−159, −37) | 348 (264, 432) | 1,519 (1,414, 1,624) |
| North West | 1,529 (1,494, 1,564) | −181 (−216, −146) | 71 (31, 111) | −87 (−144, −29) | 37 (−36, 109) | 1,550 (1,384, 1,716) |
| Yorkshire and Humber | 1,265 (1,239, 1,290) | −155 (−181, −130) | 32 (−17, 80) | −34 (−92, 24) | 85 (8, 162) | 1,347 (1,264, 1,429) |
| East Midlands | 1,074 (1,046, 1,102) | −155 (−183, −127) | 114 (70, 157) | −47 (−96, 2) | 63 (−17, 142) | 1,203 (1,089, 1,318) |
| West Midlands | 1,272 (1,245, 1,300) | −122 (−150, −95) | 163 (112, 213) | −61 (−109, −13) | 101 (42, 161) | 1,475 (1,324, 1,626) |
| East of England | 840 (814, 865) | −135 (−160, −109) | 13 (−25, 52) | −153 (−214, −93) | 80 (−15, 175) | 779 (644, 914) |
| London | 1,294 (1,266, 1,321) | −19 (−47, 9) | 117 (83, 151) | −67 (−110, −25) | 68 (25, 110) | 1,411 (1,300, 1,521) |
| South East Coast | 808 (782, 833) | −146 (−172, −121) | 64 (16, 113) | −127 (−186, −68) | 187 (124, 250) | 932 (802, 1,062) |
| South Central | 699 (673, 726) | −90 (−117, −64) | 99 (53, 144) | −74 (−113, −35) | 11 (−66, 88) | 735 (593, 877) |
| South West | 438 (408, 468) | −160 (−190, −130) | 65 (9, 122) | −31 (−95, 33) | 18 (−44, 80) | 490 (319, 661) |
| Wales | 1,014 (986, 1,042) | −284 (−311, −256) | 123 (64, 182) | −35 (−83, 12) | −115 (−178, −53) | 986 (871, 1,102) |
| *Deprivation quintiles* | | | | | | |
| 1 (least deprived) | 789 (770, 809) | −92 (−111, −72) | 79 (47, 111) | −21 (−59, 17) | 69 (−2, 139) | 916 (820, 1,012) |
| 2 | 913 (888, 938) | −116 (−141, −91) | 23 (−20, 65) | −85 (−133, −37) | 74 (18, 130) | 925 (804, 1,045) |
| 3 | 952 (929, 974) | −147 (−169, −124) | 74 (23, 124) | −116 (−161, −70) | 68 (12, 124) | 977 (869, 1,085) |
| 4 | 1,163 (1,137, 1,189) | −177 (−202, −151) | 83 (43, 123) | −83 (−140, −26) | 55 (7, 102) | 1,218 (1,089, 1,346) |
| 5 (most deprived) | 1,493 (1,459, 1,527) | −192 (−226, −158) | 144 (97, 191) | −72 (−138, −5) | 80 (21, 139) | 1,645 (1,472, 1,819) |

YLL, years of life lost.

among estimated excess respiratory deaths, a mean of 8.9 years per death (95% CI: 8.7 to 9.1) were lost in the least deprived quintile, compared to 11.2 (95% CI: 11.0 to 11.5) in the most deprived. London had the highest rate of excess deaths in people aged 74 or below, followed by the North West (Table F in S1 Appendix).

## Cardiovascular disease and diabetes

We observed an increase in YLL due to cardiovascular and diabetes deaths, with an estimated 65,941 (95% CI: 47,542 to 84,340) excess YLL nationally, reflecting a 6% (95% CI: 4% to 7%) increase compared to the equivalent time period in 2019. Furthermore, YLL attributable to cardiovascular and diabetes deaths was over 3 times greater for males, compared to females (50,877; 95% CI: 43,841 to 57,913 versus 15,064; 95% CI: 2,610 to 27,519). There was an overall

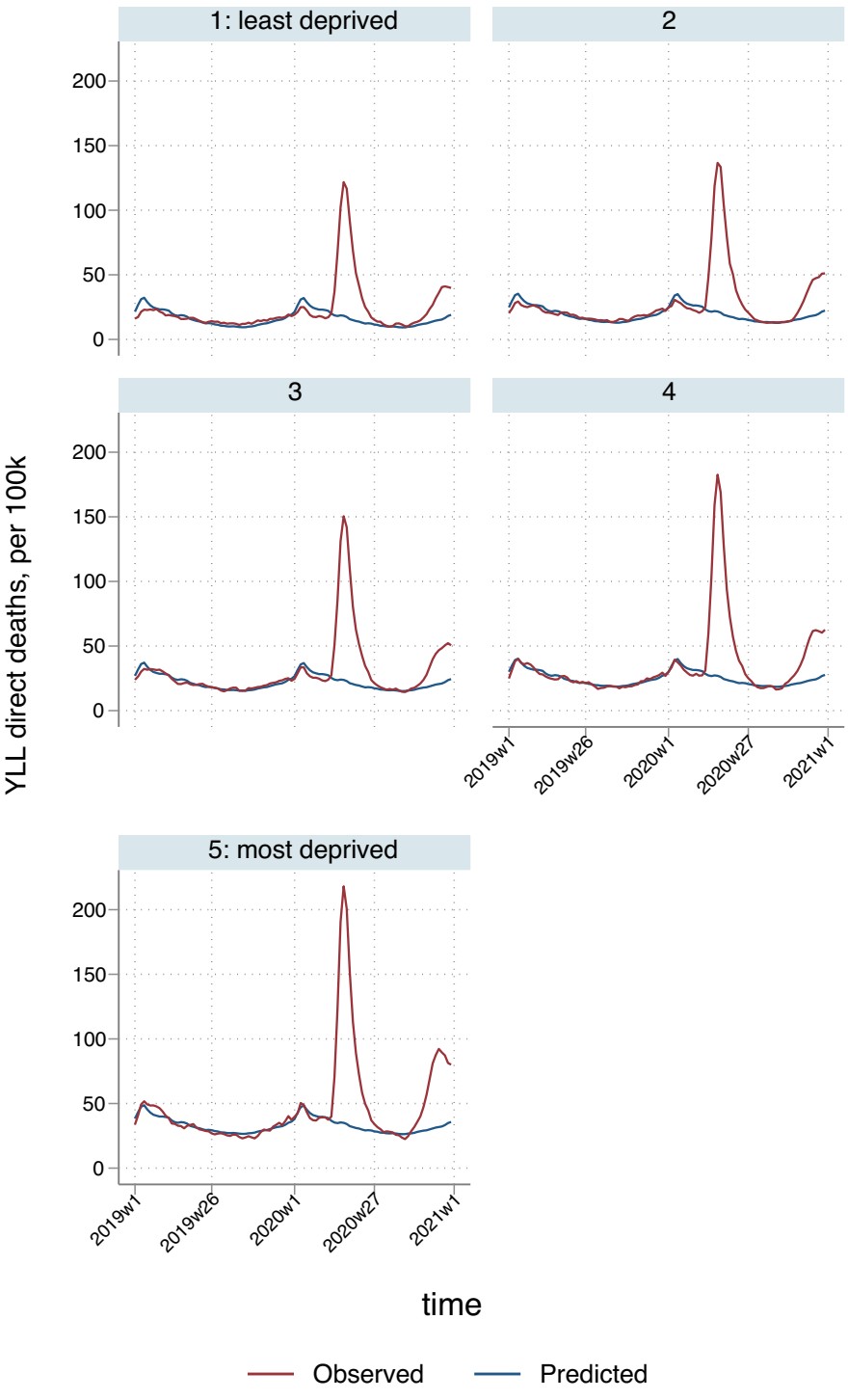

**Fig 4. Temporal patterns in YLL per 100,000 population due to direct deaths, by deprivation quintile**\*. \*3-week smoothed. YLL, years of life lost.

socioeconomic gradient evident in these excess YLL; 11,927 (95% CI: 8,994 to 14,950) and 22,776 (95% CI: 18,788 to 26,764) excess YLL were estimated in the least and most deprived areas, respectively.

Nationally, most of the excess YLL due to cardiovascular disease/diabetes occurred during the first half of the first wave of the pandemic (April to mid-May), but observed YLL exceeded predictions until the end of 2020 (Fig 5). There were 75 (95% CI: 31 to 119) excess YLL per

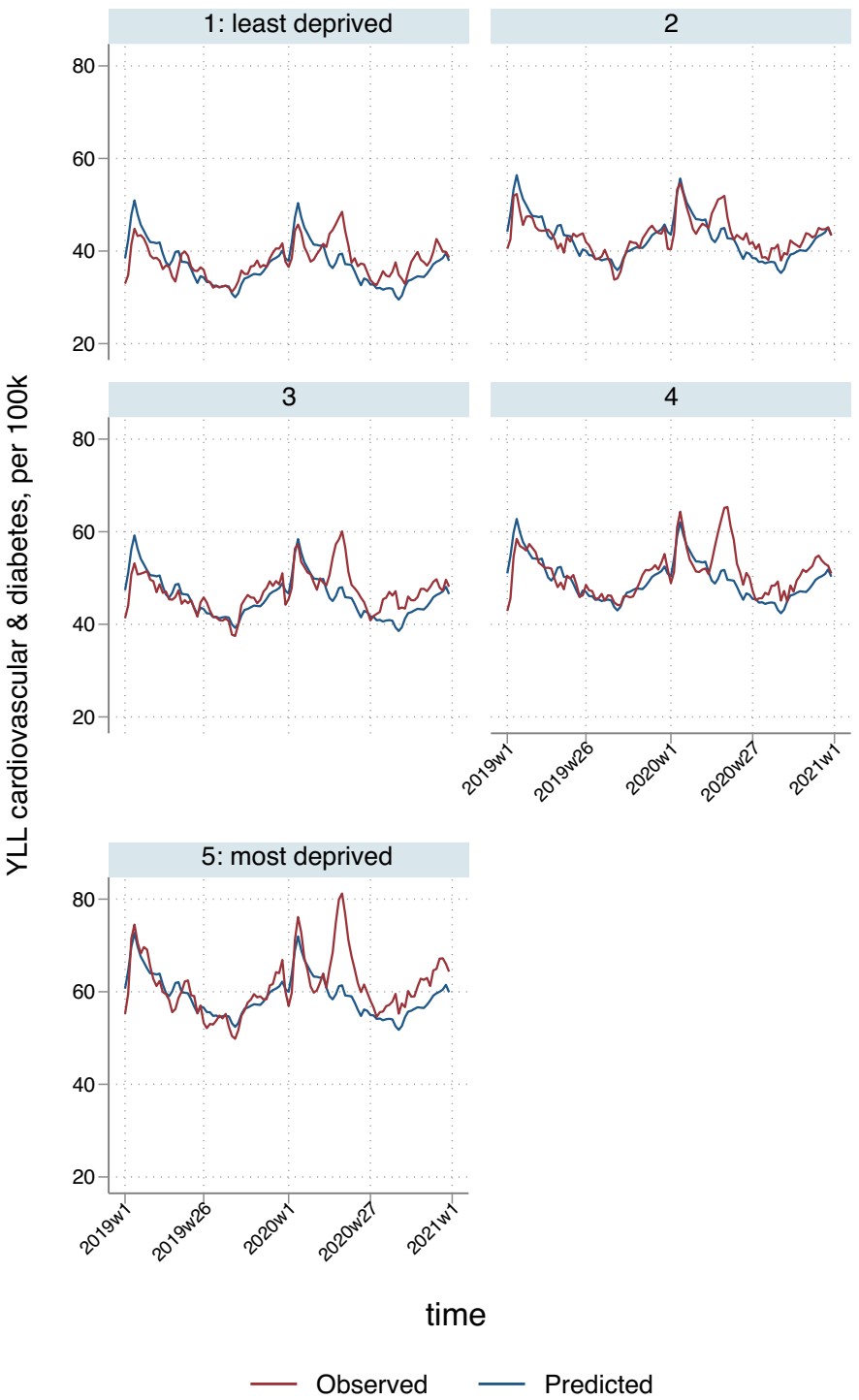

**Fig 5. Temporal patterns in YLL per 100,000 population due to cardiovascular and diabetes deaths, by deprivation quintile\*.** \*3-week smoothed. YLL, years of life lost.

100,000 of the population, ranging from 13 (95% CI: −25 to 55) per 100,000 in the East of England to 163 (95% CI: 112 to 213) per 100,000 in the West Midlands. Rates ranged from 23 (95% CI: −20 to 65) per 100,000 in the second least deprived to 144 (95% CI: 97 to 191) per 100,000 in the most deprived quintile.

### Cancer and other indirect deaths (including drug-related, alcohol-specific, suicides, fatal accidents, and all other causes)

We observed no significant pandemic-related changes in YLL due to cancer. Nationally, there were −16,738 (95% CI: −39,261 to 5,785) excess YLL due to cancer deaths over the study period. Regional and deprivation quintile estimates were small and, in some cases, not statistically significant. The highest rates per 100,000 population were in the least deprived quintile and in the South West region, but even here estimates were negative, albeit nonsignificant.

Nationally, there were 67,828 (95% CI: 25,952 to 109,705) excess YLL attributable to other indirect deaths, with wide regional variability, from 12,653 (95% CI: 10,181 to 15,126) in the North East to −608 (95% CI: −2,864 to 1,648) in Wales. Excess YLL ranged from 11,710 (95% CI: 2,694 to 20,725) in the least deprived quintile to 18,298 (95% CI: 10,754 to 25,810) in the most deprived. Estimates of rates of excess YLL were nonsignificant for some regions, with substantially higher rates in the North East of England (348 per 100,000). There was no clear socioeconomic pattern, although the highest YLL excess rates were observed in the most deprived quintile (80 per 100,000).

Table A in S1 Appendix presents total observed YLL across each category, and Table B in S1 Appendix presents and percentages of excess YLL in relation to these totals. In S1 Appendix, we also provide YLL denominators for the equivalent weeks in 2019 (Table C in S1 Appendix), and the percentages of excess YLL in 2020, in relation to the 2019 baselines (Table D in S1 Appendix). Persistent socioeconomic and regional health inequalities exist in England and Wales, as indicated by YLL. That is, in the last 42 weeks of 2019 (the equivalent time period we investigated in 2020 to quantify the impact of the COVID-19 pandemic), there was a steep socioeconomic gradient in YLL, with 804,312 in the least deprived quintile, increased to 1,255,588 in the most deprived. The ratio of YLL in the most deprived over the least deprived quintile was 1.56. For the equivalent time period in 2020, this ratio increased to 1.64 (1,501,323/913,418).

## Discussion

Before the COVID-19 pandemic, there were clear and long-standing socioeconomic and geographic patterns in YLL in England and Wales, such that years lost increased with material deprivation and were persistently higher in northern English regions and in Wales, especially for males [23,24]. This is reflected in prepandemic differences in the ASMRs across deprivation quintiles (Fig 1 and S1 Appendix). In the first 42 weeks of the pandemic, these underlying inequalities widened as the most deprived areas experienced the greatest losses, with total excess YLL nearly twice as high in the most deprived quintile, compared to the most affluent. Similarly, the most deprived areas experienced the greatest proportional increase in excess YLL, compared with the least deprived. Higher rates of YLL occurred in northern regions, although the West Midlands and London were also badly affected, most likely reflecting poverty in the major conurbations. Patterns for indirect deaths, i.e., those not directly attributed to COVID-19, varied by cause. Excess YLL from deaths attributable to cardiovascular disease and diabetes followed a similar socioeconomic pattern, and no obvious pattern was apparent for other deaths.

Our analysis provides a comprehensive picture of excess YLL during the first 42 weeks of pandemic, including major causes both related and unrelated to COVID-19 infection. This enabled us to estimate the total impact of the pandemic on YLL, including regional and area-level socioeconomic patterns. There are some limitations to our study. For deaths not referred for investigation and adjudication by a coroner, data on cause of death rely on accurate diagnosis and recording by clinicians. During a pandemic, assessing the contribution of COVID-19 to some deaths can be challenging [25]. We defined COVID-19 deaths as those for which the underlying cause was attributed to the virus on the death certificate. Inevitably, there will have been some misclassification, particularly in the early part of the pandemic when COVID-19 testing was not widespread and doctors' awareness may have been more limited [26, 27]. This is also problematic for estimating YLL by deaths caused indirectly by the pandemic, and so respiratory causes of deaths, the most likely source of "missed" COVID-19 diagnoses, were grouped together with COVID-19 deaths. Furthermore, this approach enabled us to apply data from earlier years to create a reference "baseline", in the absence of COVID-19 deaths occurring before 2020. Consequently, our estimates of excess YLL that occurred as an indirect effect of the pandemic do not include respiratory deaths not attributable to COVID-19 and are unlikely to include substantial numbers caused directly by COVID-19 infection. The aggregates of weekly estimates that we have reported included negative estimates (i.e., they were not set to zero), to allow for more conservative estimates. We did not control for population trends, assuming small increases over time, that might have a small impact on our estimates. However, in our previous work, we found that controlling for (estimated) population trends had minimal impact [2]. There was also relatively high variability in the weekly time trends of some subgroups, due to relatively small numbers, but that uncertainty is reflected in the estimated confidence intervals. The survival tables we used were not region specific, to highlight baseline differences across regions, and the choice of survival tables, national or regional, would not have greatly affected our estimates of excess YLL since they are based on historical trends. To obtain a YLL estimate per death, we assumed that all excess YLLs occurred in the excess deaths, when that is not necessarily the case—some YLLs may have occurred in "baseline" deaths. Finally, time lags for coroners' verdicts will not have been sufficiently long for many cases, particularly for external causes of death such as accidents and suicides, and we are therefore unable to determine excess YLL specific to these categories. We categorised groups to minimise the impact of these time lags, with the most affected causes contributing to a single group along with those whose causes remained undetermined ("other indirect deaths").

We estimate that there were 763,550 excess YLL during 2020 in England and Wales since the start of the pandemic, of which 646,518 (84.7%) were directly attributed to COVID-19 or other respiratory diseases, and 117,032 (15.3%) were attributed to other causes. The greatest contributors to indirect deaths were cardiovascular disease and diabetes (65,941 YLL), with three-quarters of these lost years occurring in males. This may partially reflect undiagnosed COVID-19 infection, but it is also consistent with reduced capacity for investigation and elective treatment of cardiovascular disease and hospital avoidance for acute cardiovascular events [3]. Our observations around the relatively small negative estimates for YLL as a result of cancer may relate to decreases in the diagnosis of cancer during the pandemic, where patients were reluctant to seek medical advice for cancer symptoms and elective investigations (particularly cancer imaging) were delayed to protect patients and staff from exposure to COVID-19, and therefore undiagnosed cancer deaths were assigned to other causes [28]. In addition, the impact of the pandemic on cancer mortality is likely to be longer term. We have previously estimated that 26% of excess deaths during the pandemic were attributable to nonrespiratory causes [2], which suggests that these indirect deaths had an older age profile than deaths directly attributed to COVID-19, and therefore comprised a lower proportion of the total YLL.

The socioeconomic gradient in total YLL was steeper than that previously found for excess deaths [2]. Over 50% of excess YLL were in the 2 most deprived quintiles, indicating that a disproportionate number of deaths in these areas, of which there were more than in more affluent areas, were in younger age groups. Given that the most deprived parts of the country had a far higher prepandemic level of YLL, this means that the already wide health inequalities in England and Wales were magnified during the 2020 phase of the COVID-19 pandemic, in both absolute and relative terms. In our previous work, we observed that in the first 30 weeks of the pandemic mortality rates in the most deprived quintile were 1.13 (72,595/64,109) times that in the least [2]. However, for the first 42 weeks of the pandemic, we found that YLL were 1.64 higher (1,501,323/913,418; Table A in S1 Appendix), indicating that a higher proportion of total deaths were in younger age groups in more deprived areas. Similarly, regions of England and Wales with persistently high YLL such as the North East and North West of England had the greatest increases in YLL. For the North East, rates of excess direct YLL were substantially lower than for the North West, and rates of "other" YLL were far higher, suggesting that there may have been lower rates of recording of COVID-19 on the death certificates of younger people in the North East. However, factors other than deprivation will also have contributed, with relatively high rates of YLL in London and the West Midlands and relatively low rates in the South West and Wales, given these regions' health, ethnicity, and deprivation profiles.

The COVID-19 pandemic has widened preexisting health inequalities across England and Wales: Regions and social groups with the highest baseline mortality rates experienced the greatest excesses in YLL. Our findings support the notion that YLL can be more informative for determining unmet needs and are in line with several of the recommendations made by the British Medical Association for mitigating the effects of COVID-19. These include prioritising vaccine delivery and providing targeted financial and social support for groups severely impacted by both the pandemic disease and the public health measures implemented to reduce transmission [29], as well as increased support for primary care services in the areas most affected. Many of these groups were already compromised in financial and health terms by the 2007–2009 Great Recession and subsequent years of austerity, and face further economic uncertainty following the pandemic [30]. Successive economic shocks place constraints on policymakers, but if health inequalities are to be attenuated, and ideally significantly reduced, a robust policy response is required, including substantial investment in future pandemic preparedness and to support chronically stretched health services [31]. Finally, more effective public health actions must continue at the national level to address the full range of socioeconomic determinants of ill health.

## Supporting information

**S1 Checklist. STROBE checklist.**
(PDF)

**S1 Appendix. Supporting information providing additional information on denominators and figures, by each categorisation of interest.**
(PDF)

**S1 File. Stata commands: Caller for all other programmes.**
(DO)

**S2 File. Stata commands: Analyses on excess mortality at the aggregate level: England and Wales.**
(DO)

**S3 File. Stata commands: Analyses on excess mortality in deprivation strata.**
(DO)

**S4 File. Stata commands: Analyses on excess mortality in region strata.**
(DO)

**S5 File. Stata commands: Analyses on excess mortality in region by deprivation strata.**
(DO)

**S6 File. Stata commands: Analyses on excess mortality in sex strata.**
(DO)

**S7 File. Stata commands: All excess mortality results aggregated in a table.**
(DO)

**S8 File. Stata commands: Mortality denominators reported in tables.**
(DO)

**S9 File. Stata commands: Analyses on excess mortality per 100,000 population at the aggregate level: England and Wales.**
(DO)

**S10 File. Stata commands: Analyses on excess mortality per 100,000 population in deprivation strata.**
(DO)

**S11 File. Stata commands: Analyses on excess mortality per 100,000 population in region strata.**
(DO)

**S12 File. Stata commands: Analyses on excess mortality per 100,000 population in region by deprivation strata.**
(DO)

**S13 File. Stata commands: Analyses on excess mortality per 100,000 population in sex strata.**
(DO)

**S14 File. Stata commands: All excess mortality per 100,000 population results aggregated in a table.**
(DO)

## Author Contributions

**Conceptualization:** Evangelos Kontopantelis, Mamas A. Mamas, Roger T. Webb, Martin K. Rutter, Darren M. Ashcroft, Matthias Pierce, Kathryn M. Abel, Gareth Price, Corinne Faivre-Finn, Harriette G. C. Van Spall, Michelle M. Graham, Marcello Morciano, Glen P. Martin, Matt Sutton, Tim Doran.

**Data curation:** Evangelos Kontopantelis.

**Formal analysis:** Evangelos Kontopantelis.

**Investigation:** Evangelos Kontopantelis, Tim Doran.

**Methodology:** Evangelos Kontopantelis, Tim Doran.

**Project administration:** Evangelos Kontopantelis, Mamas A. Mamas, Ana Castro.

**Resources:** Mamas A. Mamas.

**Software:** Evangelos Kontopantelis.

**Validation:** Glen P. Martin.

**Visualization:** Evangelos Kontopantelis.

**Writing – original draft:** Evangelos Kontopantelis, Tim Doran.

**Writing – review & editing:** Evangelos Kontopantelis, Mamas A. Mamas, Roger T. Webb, Ana Castro, Martin K. Rutter, Chris P. Gale, Darren M. Ashcroft, Matthias Pierce, Kathryn M. Abel, Gareth Price, Corinne Faivre-Finn, Harriette G. C. Van Spall, Michelle M. Graham, Marcello Morciano, Glen P. Martin, Matt Sutton.

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
