## [Editor Report · Decision Letter 0]

3 Aug 2021

Dear Dr Kontopantelis, 

Thank you for submitting your manuscript entitled "Excess years of life lost to COVID-19 and other causes of death by sex, neighbourhood deprivation and region in England & Wales during 2020" for consideration by PLOS Medicine.

Your manuscript has now been evaluated by the PLOS Medicine editorial staff and I am writing to let you know that we would like to send your submission out for external peer review.

Please re-submit your manuscript within two working days, i.e. by Aug 05 2021 11:59PM.

Kind regards,

Louise Gaynor-Brook, MBBS PhD

Associate Editor

PLOS Medicine

---

## [Decision Letter · Decision Letter 1]

29 Sep 2021

Dear Prof. Kontopantelis,

Thank you very much for submitting your manuscript "Excess years of life lost to COVID-19 and other causes of death by sex, neighbourhood deprivation and region in England & Wales during 2020" (PMEDICINE-D-21-03320R1) for consideration at PLOS Medicine. 

Your paper was evaluated by three independent reviewers, including a statistical reviewer, and was discussed among all the editors here and with an academic editor with relevant expertise. The reviews are appended at the bottom of this email and any accompanying reviewer attachments can be seen via the link below:

[LINK]

In light of these reviews, I am afraid that we will not be able to accept the manuscript for publication in the journal in its current form, but we would like to consider a revised version that addresses the reviewers' and editors' comments. Obviously we cannot make any decision about publication until we have seen the revised manuscript and your response, and we plan to seek re-review by one or more of the reviewers. 

We expect to receive your revised manuscript by Oct 20 2021 11:59PM. Please email us (plosmedicine@plos.org) if you have any questions or concerns.

We look forward to receiving your revised manuscript. 

Sincerely,

Louise Gaynor-Brook, MBBS PhD

Associate Editor, PLOS Medicine

plosmedicine.org

General comments:

Please provide line numbers in your revised manuscript, ideally not beginning from 1 with each new page.

Throughout the paper, please adapt reference call-outs to the following style: "... decisions on seeking care [5,6]." (noting the absence of spaces within the square brackets).

Data availability:

PLOS Medicine requires that the de-identified data underlying the specific results in a published article be made available, without restrictions on access, in a public repository or as Supporting Information at the time of article publication, provided it is legal and ethical to do so. 

Since some of the data are not freely available, please describe briefly the ethical, legal, or contractual restriction that prevents you from sharing it. Please also include an appropriate contact (web or email address) for inquiries - please note that a study author cannot be the contact person for the data. 

Title: Please revise your title according to PLOS Medicine's style. Please place the study design in the subtitle (ie, after a colon). We suggest “Excess years of life lost to COVID-19 and other causes of death by sex, neighbourhood deprivation and region in England & Wales during 2020: A registry-based study” or similar

Abstract:

Please structure your abstract using the PLOS Medicine headings (Background, Methods and Findings, Conclusions), combining Methods and Findings sections into one section. 

Abstract Background: Please provide the context of why the study is important. The final sentence should clearly state the study question.

Abstract Methods and Findings:

Please provide brief demographic details of the study population (e.g. sex, age, ethnicity, etc)

Please include the study design, number of individuals included, and main outcome measures.

Please comment on other findings from the study relating to age and region

In the last sentence of the Abstract Methods and Findings section, please describe 2-3 of the main limitations of the study's methodology.

Abstract Conclusions:

Please begin your Abstract Conclusions with "In this study, we observed ..." or similar, to summarize the main findings from your study, without overstating your conclusions. Please emphasize what is new and address the implications of your study, being careful to avoid assertions of primacy. 

Author Summary:

In the final bullet point of ‘What Do These Findings Mean?’, please describe the main limitations of the study in non-technical language.

Introduction:

Please indicate whether your study is novel, being careful to temper assertions of primacy by adding ‘to the best of our knowledge’ or similar. 

The terms gender and sex are not interchangeable (as discussed in http://www.who.int/gender/whatisgender/en/ ); please use the appropriate term.

Please conclude the Introduction with a clear description of the study question or hypothesis.

Methods:

Did your study have a prospective protocol or analysis plan? Please state this (either way) early in the Methods section. If a prospective analysis plan (from your funding proposal, IRB or other ethics committee submission, study protocol, or other planning document written before analyzing the data) was used in designing the study, please include the relevant prospectively written document with your revised manuscript as a Supporting Information file to be published alongside your study, and cite it in the Methods section. A legend for this file should be included at the end of your manuscript. If no such document exists, please make sure that the Methods section transparently describes when analyses were planned, and if/when reported analyses differed from those that were planned. Changes in the analysis-- including those made in response to peer review comments-- should be identified as such in the Methods section of the paper, with rationale. If a reported analysis was performed based on an interesting but unanticipated pattern in the data, please be clear that the analysis was data-driven.

Ethics statement: Please state in the methods why your study did not require an ethics statement.

Please ensure that the study is reported according to the RECORD guideline, and include the completed RECORD checklist as Supporting Information. Please add the following statement, or similar, to the Methods: "This study is reported as per the REporting of studies Conducted using Observational Routinely-collected Data (RECORD) guideline (S1 Checklist)." The RECORD guideline can be found here: https://www.record-statement.org/ When completing the checklist, please use section and paragraph numbers, rather than page numbers which will likely no longer correspond to the appropriate sections after copy-editing.

“More details on the selected groups have been provided elsewhere” - please provide more details in the supplementary materials

There is an unusually high degree of matching between your methods section and that of reference 2; please address this in your revision via rephrasing throughout the relevant section

Results: 

Please provide a table showing the baseline characteristics of the study population (Table 1).

Please report the number of individuals included in your analysis. 

Please refer to the relevant figures and tables to support the findings presented in the main text. 

Discussion:

Please present and organize the Discussion as follows: a short, clear summary of the article's findings; what the study adds to existing research and where and why the results may differ from previous research; strengths and limitations of the study; implications and next steps for research, clinical practice, and/or public policy; one-paragraph conclusion.

Please remove all subheadings within your Discussion 

Figures:

Please provide titles and legends for each individual figure 

Tables:

Please provide titles and legends for each individual table 

Tables 1, 2 Please clarify what the numbers in brackets represent in the respective table legends

References:

Please ensure that journal name abbreviations match those found in the National Center for Biotechnology Information (NCBI) databases, and are appropriately formatted and capitalised.

Please also see https://journals.plos.org/plosmedicine/s/submission-guidelines#loc-references for further details on reference formatting. 

Where website addresses are cited, please specify the date of access. 

Supplementary files: 

Please provide titles and legends for each individual table and figure in the Supporting Information.

Please see https://journals.plos.org/plosmedicine/s/supporting-information for our supporting information guidelines. 

Comments from the reviewers:

Reviewer #1: The manuscript describes an interesting study, that is very relevant for both estimating the overall impact of COVID-19 on excess mortality in the UK, and more specifically its impact on socioeconomic and regional inequalities. The use of the years of life lost is fairly novel, and adds to the relevance of the work, as it makes it easier to interpret. Additionally, this takes into account the differential impact of COVID-19 by age groups. The paper is well written and generally provides enough information on the various steps taken in the analyses. However, I do have some issues that I would like to see addressed, detailed below.

Major issues

I find the classification of direct and indirect excess death somewhat problematic. Including non-COVID-19 specific respiratory deaths as direct COVID-19 deaths due to potential misclassification is understandable, and addressed in the limitations section. However, the inclusion of indirect deaths could use more elaboration, both in the introduction section (to justify including it), and in the methods section (how it is defined). That deaths due to cardiovascular disease and diabetes may be related to COVID-19 seems somewhat plausible. Although as far as I know, these are risk factors that make a person more susceptible to (severe outcomes) of COVID-19, which does not necessarily mean that dying from cardiovascular disease or diabetes is indirectly caused by COVID-19. How cancer deaths and all other deaths (including suicides and accidents) are counted as indirectly caused by COVID-19, I find harder to understand. Perhaps, it would make more sense to simply split the results into direct COVID-19 deaths, and all-cause deaths. Or use different terminology, as in the discussion, the authors both mention "indirect deaths, i.e. those not directly attributed to COVID-19", and causes both related and unrelated to COVID-19 infection. I personally think that unrelated seems to be more fitting for most of the categories that are currently defined as indirect.

Minor issues

- In the abstract and methods, the authors mention that data is obtained from the end of 2014 onwards, however they also mention that 2019 is used as the baseline period to which the 2020 data are compared. It is not clear to me whether data from 2014 to 2018 were used and if so, why? Please clarify which data period was used as the reference.

- In calculating the index of multiple deprivation, data were used for England from 2019, and for Wales from 2014. Does this not create some problem in comparing the two? The authors do state that these scores are relatively constant over time, would it then not make more sense to use data from 2014 for both countries?

- In presenting the results, the authors first describe the years of life lost in absolute numbers, and then again, standardized per 100,000 people. The former might be relevant for policymakers, or people very familiar with the total number of people living in the various described regions. However, for the readership of a scientific journal such as this, it seems somewhat redundant, and the numbers per 100,000 are much easier to interpret and compare.

- In the results section, heading Respiratory diseases, the authors mention the number of YLL and state that this is 'an 123% increase … compared to the equivalent time period in 2019'. This comparison sound problematic to me, as there were 0 COVID-19 related deaths in 2019. If this is comparing all respiratory deaths to those of 2019, this should be specified more clearly.

- In the sentence: "Excess respiratory deaths across age-groups, … in younger age groups (45-64: 2008/681; 65-74: 2774/1377; 75-84: 5069/3655; 85+: 5302/5636).", these ratios are not intuitive to interpret. It would be better to write these out as in done in the last paragraph of the results section, e.g. "ratio increased to 1.64 (1,501,323/913,418)."

- "Assuming estimated excess respiratory YLL were only distributed amongst estimated excess respiratory deaths,…". It is unclear what is meant in this sentence, please reformulate it to improve clarity.

- In the discussion, the authors mention that "in addition to excluding COVID-19 deaths from the total excess deaths, we also excluded other respiratory causes of death (the most likely source of 'missed' COVID-19 diagnoses)." Firstly, I think this should be mentioned under the methods section, and not only in the discussion section. Secondly, I understand that both COVID-19 deaths and respiratory deaths might be liable to 'missed' diagnoses, but in my opinion, that should not matter for reporting total excess deaths. When one read total excess deaths, one assumes this includes all causes, regardless of whether they were correctly classified or not.

- The authors state: "We did not control for population trends, assuming small increases over time …". Referring back to my point about which time period was used as comparison, if only 2019 data was used as baseline, then I agree that controlling for population trends is probably not necessary. However, if data from 2014 onwards was used, I would find this assumption more problematic, and would want to see some description of existing trends as justification for (not) controlling for them.

- Tables 1 and 2 in their current form are very hard to read. Adding thousands separators would help, Moreover, there is no table legend to clarify which numbers are presented, and which are in parentheses. Also, referring back to my earlier comment, I think mostly table 2 with the results per 100,000 are useful for a broad audience.

Reviewer #2: "Excess years of life lost to COVID-19 and other causes of death by sex, neighbourhood deprivation and region in England & Wales during 2020" focuses on the (excess) years of life lost (YLL) metric, for assessing mortality due to the coronavirus either directly or indirectly, in England and Wales across various sex, geographical region and deprivation demographics. For the period approximately between March and December 2020, a 15% excess YLL was found, with relatively more excess YLL amongst the more-deprived, and also for younger age groups across deprivation quintiles. Analyses were performed for all-cause, respiratory disease-based, cardiovascular disease/diabetes-based and cancer/other indirect deaths. It should be noted that deprivation was estimated based on residence by postcode, and not from actual individual information.

The results are largely devoted to describing observed trends by group category, in detail. The paper appears of potential significance in quantifying the burden of COVID-19 on relatively-deprived neighbourhoods/regions. However, a number of issues might be addressed:

1. It might be considered to emphasize the additional contributions of this manuscript over prior related work by much the same authors (cited as [2]), in the Introduction.

2. It is stated that YLLs are calculated against 2019 life expectancy (79.4 years for males, 83.1 years for females), and that "Any death over those age thresholds, for males and females respectively does not contribute to the YLL total for the particular stratum" (Appendix Section 1). While this appears a possible definition for YLL, a perhaps more appropriate definition for YLL would involve the difference between age at death, and the standard life expectancy (SLE) at that age (see for example "Reflection on modern methods: years of life lost due to premature mortality—a versatile and comprehensive measure for monitoring non-communicable disease mortality", Martinez et al., International Journal of Epidemiology 48:4). For example, if a male aged 80 years perishes, then YLL would be (SLE at 80)-80, instead of zero as appears to currently be the case. This would seem more accurate since the (possibly-mitigable) death of subjects above the (average) life expectancy should intuitively result in some appreciable YLL instead of having no impact as currently defined, and would appear to underplay the impact on deaths amongst the elderly, who have been widely recognized as being the age group most vulnerable to the coronavirus.

If the above is correct, it might therefore be considered to perform some sensitivity analyses taking SLE into consideration, and also to clarify the statement that "Those aged over 100 were assumed to have the life expectancy of centenarians of the same sex", since it is not immediately clear whether they contribute to the YLL total in the first place, from the Appendix definition.

3. Conventions for ICD-10 coding might be briefly clarified. Might it be possible to have a death attributed to both direct and indirect causes (relative to COVID-19)? If so, how would they be accounted for, e.g. direct cause taking precedence, or under both causes?

4. "Maximum lag order" might be briefly clarified, and the significance of the chosen value (52 weeks) explained.

5. The correspondence between quintile number and deprivation order might be stated, at the first mention of quintile number.

6. It might be interesting to briefly note any interaction effects between the major categories, e.g. for generally more-deprived regions, is the increased YLL ratio effect for more-deprived quintiles amplified, constant or attenuated?

7. It might be considered to provide an example of the projected YLL vs. observed annual YLL for past registered deaths (i.e. for each year from 2015 onwards) on a single chart, towards illustrating/explaining expected YLL. Moreover, if possible, if might be considered to demonstrate the validity of this procedure, by also displaying the expected YLL for say 2019, as estimated by data from 2015-2018.

8. In general, it would be strongly suggested to annotate/caption Appendix charts, on the same page as the chart itself. For example, under Section 5.3.4, groups of charts with the same axes (time/YLL grand total, per 100k) and regions (North East, North West, etc.) are displayed on consecutive pages (Page 47 to 50) without further indication as to what they represent. This has made evaluation of the Appendix data challenging. 

Reviewer #3: This research shows the behavior of excess YLL in England & Wales and how it was affected by socioeconomic conditions reflecting important inequities exacerbated of the SARS-CoV-2 pandemic. The article is well written and methodologically correct. Despite this, I wish to make a few recommendations that may be helpful to readers, particularly non-England & Wales readers. In the first place, it would be interesting to mention in the discussion what is the percentage of underreporting of mortality in England & Wales and to what extent this could affect the estimates of excess YLL. Second, that comparisons are made within England & Wales, it would be important to include in the discussion comparisons of the results with other European countries, in order to understand if the impact of the pandemic on the excess of YLL in England & Wales was higher or lower than other countries. A map showing the regions of England & Wales selected for analysis would be appreciated.

[LINK]

---

## [Decision Letter · Decision Letter 2]

17 Nov 2021

Dear Dr. Kontopantelis,

Thank you very much for re-submitting your manuscript "Excess years of life lost to COVID-19 and other causes of death by sex, neighbourhood deprivation and region in England & Wales during 2020: a registry-based study" (PMEDICINE-D-21-03320R2) for review by PLOS Medicine.

I have discussed the paper with my colleagues and the academic editor and it was also seen again by three reviewers. I am pleased to say that provided the remaining editorial and production issues are dealt with we are planning to accept the paper for publication in the journal.

[LINK]

We look forward to receiving the revised manuscript by Nov 24 2021 11:59PM.   

Sincerely,

Beryne Odeny (for Louise Gaynor-Brook, MBBS PhD)

PLOS Medicine

plosmedicine.org

Requests from Editors:

1. Data availability: PLOS Medicine requires a “minimal data set” which consists of the data set used to reach the conclusions drawn in the manuscript with related metadata and methods, and any additional data required to replicate the reported study findings in their entirety. Authors do not need to submit their entire data set, or the raw data collected during an investigation. Please submit the following data: The values behind the means, standard deviations and other measures reported; The values used to build graphs; The points extracted from images for analysis. 

2. Please relocate the limitations in the Abstract to the end of the Methods and Findings section

3. Pleas avoid use of double negatives such as “losses in … life lost” (line 110 in the Abstract)

4. In the conclusion (abstract and main text), please focus on reporting what you have observed and the implications, rather than trying to conclude that “inequalities … were exacerbated"

5. Please add the following statement, or similar, to the Methods: "This study is reported as per the Strengthening the Reporting of Observational Studies in Epidemiology (STROBE) guideline (S1 Checklist)."

Comments from Reviewers:

Reviewer #1: My previous comments have mostly been sufficiently addressed. I maintain my point that Table 1 (and 2) is hard to read in its current form, but I agree that that could be further addressed in the production stage.

Reviewer #2: We thank the authors for addressing our previous concerns.

Reviewer #3: I consider that the authors have adequately resolved the comments raised by the reviewers. More methodological detail is now observed and the limitations of the study as well as potential biases have been adequately addressed. In this way, the results better reflect the existing sociodemographic inequities in England and Walles and how these have been exacerbated by the pandemic.

Willy Ramos

[LINK]

---

## [Editor Report · Decision Letter 3]

5 Jan 2022

Dear Dr Kontopantelis, 

On behalf of my colleagues and the Academic Editor, Prof. Elvin Geng, I am pleased to inform you that we have agreed to publish your manuscript "Excess years of life lost to COVID-19 and other causes of death by sex, neighbourhood deprivation and region in England & Wales during 2020: a registry-based study" (PMEDICINE-D-21-03320R3) in PLOS Medicine.

Before your manuscript can be formally accepted you will need to complete some formatting changes and some final editorial requests, which you will receive in a follow up email. Please be aware that it may take several days for you to receive this email; during this time no action is required by you. Once you have received these formatting requests, please note that your manuscript will not be scheduled for publication until you have made the required changes. 

PRESS

Sincerely, 

Louise Gaynor-Brook, MBBS PhD 

Associate Editor, PLOS Medicine